# Perioperative morbidity and mortality of cardiothoracic surgery in patients with a do-not-resuscitate order

Bryan G. Maxwell[1], Robert L. Lobato[2], Molly B. Cason[1] and Jim K. Wong[3]

[1] Department of Anesthesiology and Critical Care Medicine, Johns Hopkins University School of Medicine, Baltimore, MD, USA
[2] Department of Anesthesia, Cedars-Sinai Medical Center, Los Angeles, CA, USA
[3] Department of Anesthesia, Stanford University School of Medicine, Stanford, CA, USA

## ABSTRACT

**Background.** Do-not-resuscitate (DNR) orders are often active in patients with multiple comorbidities and a short natural life expectancy, but limited information exists as to how often these patients undergo high-risk operations and of the perioperative outcomes in this population.

**Methods.** Using comprehensive inpatient administrative data from the Public Discharge Data file (years 2005 through 2010) of the California Office of Statewide Health Planning and Development, which includes a dedicated variable recording DNR status, we identified cohorts of DNR patients who underwent major cardiac or thoracic operations and compared them to age- and procedure-matched comparison cohorts. The primary study outcome was in-hospital mortality.

**Results.** DNR status was not uncommon in cardiac ($n = 2,678$, 1.1% of all admissions for cardiac surgery, age $71.6 \pm 15.9$ years) and thoracic ($n = 3,129$, 3.7% of all admissions for thoracic surgery, age $73.8 \pm 13.6$ years) surgical patient populations. Relative to controls, patients who were DNR experienced significantly greater in-hospital mortality after cardiac (37.5% vs. 11.2%, $p < 0.0001$) and thoracic (25.4% vs. 6.4%) operations. DNR status remained an independent predictor of in-hospital mortality on multivariate analysis after adjustment for baseline and comorbid conditions in both the cardiac (OR 4.78, 95% confidence interval 4.21–5.41, $p < 0.0001$) and thoracic (OR 6.11, 95% confidence interval 5.37–6.94, $p < 0.0001$) cohorts.

**Conclusions.** DNR status is associated with worse outcomes of cardiothoracic surgery even when controlling for age, race, insurance status, and serious comorbid disease. DNR status appears to be a marker of substantial perioperative risk, and may warrant substantial consideration when framing discussions of surgical risk and benefit, resource utilization, and biomedical ethics surrounding end-of-life care.

Corresponding author
Bryan G. Maxwell,
bmaxwell@jhu.edu

## INTRODUCTION

Advance directives have an increasingly important role in contemporary medical practice, and their use has come to affect perioperative decision-making more frequently than in prior eras (*Burkle et al., 2013*). A greater attention to bioethical considerations has spurred efforts to promote the discussion and use of advance directives in the surgical setting. The American College of Surgeons[1] and the American Society of Anesthesiologists[2] have opposed the practice of requiring DNR orders to be cancelled for a patient undergoing surgery, instead recommending that DNR patients may be candidates for an operation but that their specific wishes regarding which interventions would be appropriate in the perioperative setting should be discussed in detail with the anesthesiology and surgical teams. Specific guidelines exist to detail options for individualizing care and implementing DNR orders in the perioperative setting (*Ewanchuk & Brindley, 2006*; *Truog, Waisel & Burns, 1999*; *Truog, Waisel & Burns*; *Waisel et al., 2002*).

Patient decision-making surrounding the risks and benefits of surgery and associated interventions (e.g., intubation, vasoactive medication use) in the perioperative period depend crucially on expectations about the likelihood of a successful operation, anesthetic, and postoperative recovery. However, outcome data are limited regarding the relationship between DNR status and surgical morbidity and mortality. While some studies have examined this relationship in a broad general surgical cohort (*Kazaure, Roman & Sosa, 2011*) and a colorectal surgical population (*Speicher et al., 2013*), it remains incompletely evaluated in the setting of thoracic and cardiac operations, which in general involve more complicated recovery and carry greater overall risks.

We undertook the present analysis to better evaluate the relationship between DNR status and cardiothoracic surgical outcomes. We used a large administrative dataset with prospectively entered information on DNR status to assess the hypothesis that DNR status is associated with increased perioperative morbidity and mortality.

## MATERIALS AND METHODS

The Stanford University Institutional Review Board granted an exemption from review because this research uses publicly available, deidentified data. The Public Discharge Data (PDD) file is a comprehensive public dataset of inpatient admissions consisting of one record for each patient discharge from a licensed hospital in California, provided by the California Office of Statewide Health Planning and Development (OSHPD) via the Medical Information Reporting for California (MIRCal) System. PDD records from years 2005 through 2010 were examined for this analysis.

### Cohort generation and matching

International classification of diseases, ninth revision, clinical modification (ICD-9-CM) Volume 3 procedure codes (see Appendix) were used to define a subset of admissions in which a major cardiothoracic surgical procedure was performed. DNR status was recorded from a dedicated PDD variable ("DNR") that is recorded as affirmative if a patient has a prehospital DNR order continued at the time of admission or a DNR order written within

[1] American College of Surgeons, "Statement on Advance Directives by Patients: "Do Not Resuscitate" in the Operating Room". Available at http://www.facs.org/fellows_info/statements/st-19.html. Accessed November 14, 2013.

[2] American Society of Anesthesiologists, "Ethical Guidelines for the Anesthesia Care of Patients with Do-Not-Resuscitate Orders or Other Directives That Limit Treatment (2008)". Available at http://www.asahq.org/For-Members/~/media/For%20Members/documents/Standards%20Guidelines%20Stmts/Ethical%20Guidelines%20for%20the%20Anesthesia%20Care%20of%20Patients.ashx. Accessed November 14, 2013.

24 h of hospital admission. DNR is recorded as negative if a patient has a DNR order written subsequent to the first 24 h of admission or if no active DNR order is present at any time during the admission.

Patients with a DNR order within the subset of records containing a major cardiac or thoracic operation comprised the study cohorts, and the analysis was performed separately for the cardiac and thoracic cohorts. For each cohort, matching was performed using a previously described SAS greedy caliper matching macro (*Bergstralh & Kosanke, 1995*) on the following variables: age (±1 years), sex, year of operation (exact match), and primary procedure code (exact match) to create a comparison cohort of up to 4:1 matched controls for each DNR case. Records with missing data for the matching variables were excluded. We selected a 4:1 matching ratio to optimize statistical power within reasonable limits of computational efficiency.

## Definition of comorbid and outcome variables

Comorbidity information was collected based on the combination of ICD-9-CM diagnosis codes and a PDD variable ("POA_P" and "OPOAx") recording whether any individual diagnosis code represented a condition that was present on admission (to avoid inclusion of conditions that developed during the hospitalization). The following comorbid conditions were defined: diabetes mellitus (249.x–250.x), malignancy (140.x–239.x), anemia (285.x), hypertension (401.x–405.x), chronic kidney disease (583, 584, 584.5, 584.9, 585, and 586), coronary artery disease (410.x–414.x), congestive heart failure (428.x), arrhythmia (426.x–427.x), prior cerebrovascular accident (430.x–437.x), and chronic lung disease (490.x–496.x).

The primary outcome measure was all-cause in-hospital mortality, as recorded in the PDD dataset (disposition code 11). Secondary outcome measures were defined based on the combination of ICD-9-CM diagnosis codes and a negative value for the "present on admission" variable (to avoid inclusion of conditions that were already present on admission). The following five conditions were defined as secondary outcomes: acute kidney injury (584, 584.5, 584.9, and 583), myocardial infarction (410.x–414.x), new congestive heart failure (428.x), new cerebrovascular accident (430.x–437.x), and respiratory failure (518.5 and 518.81).

## Statistical analysis

Demographic, preoperative, and perioperative outcome variables were compared between cohorts (cardiac DNR versus cardiac control, thoracic DNR versus thoracic control). Univariate analyses of the frequency of primary and secondary outcome measures were performed in the DNR compared to the control cohort. Continuous variables were compared using the Wilcoxon test. Discrete variables were compared using Fisher's exact test or Pearson's chi-squared test, as appropriate. For outcome variables, odds ratios with 95% confidence intervals also were calculated. A multivariate logistic regression model then was constructed to further evaluate the comparative effect of the presence of a DNR order on the primary outcome (in-hospital mortality) while adjusting for differences in baseline comorbid conditions that were not part of the matching algorithm. All variables

with a $p \leq 0.2$ in Table 1 were included in the multivariate model. Two- and three-way interactions between predictive variables were included for initial evaluation but retained in the final model only if statistically significant. The area under the receiver operating characteristic curve (AUC) was calculated for calibration of the models.

A predetermined alpha of 0.05 was used as the threshold of statistical significance for the primary outcome. For the purposes of evaluating the five individual secondary outcome measures, a Bonferroni-adjusted significance level of 0.01 was used to account for the increased possibility of type-I error. Analyses were performed using SAS (SAS 9.3, SAS Institute, Cary, NC, USA).

## RESULTS

Patients with an active DNR order within 24 h of admission represented 3,129 (3.7%) of 85,164 admissions for thoracic surgery and 2,678 (1.1%) of 242,234 admissions for cardiac surgery during the study period. Matching resulted in a cardiac control cohort of 10,670 admissions and a thoracic control cohort of 12,290 admissions. Demographic and comorbid characteristics of the DNR and control cohorts are shown in Table 1.

Table 2 provides a comparison of outcomes between the DNR and matched control cohorts. The primary outcome comparison revealed high in-hospital mortality in the thoracic (25.4%) and cardiac (37.5%) DNR groups that was significantly increased compared to controls ($p < 0.0001$ for both). Many but not all measures of resource utilization and secondary outcomes were worse in the DNR cohorts (see Table 2).

Multivariate logistic regression performed to evaluate the effect of DNR status while controlling for baseline differences in patient characteristics and comorbidities resulted in models with acceptable area under the ROC curve (thoracic model AUC = 0.734, cardiac model AUC = 0.711). Results are presented in Table 3. DNR status remained an independent predictor of mortality in both models ($p < 0.0001$ for both). Other independent predictors ($p$ value below the Bonferroni-adjusted significance level of 0.01) in the model for thoracic operations included arrhythmia, chronic kidney disease, and hypertension. Other independent predictors in the model for cardiac operations included coronary artery disease, congestive heart failure, chronic kidney disease, chronic lung disease, and hypertension.

## DISCUSSION

There are three primary findings of this study. First, we find it remarkable that such a substantial minority proportion of cardiothoracic surgical patients have an active DNR order in place at the time of the admission in which surgery occurs. The magnitude represented by the two study cohorts (3.7% and 1.1 of all thoracic and cardiac surgical patients, respectively) indicates that it is not an exceedingly rare event for DNR patients to be offered – and to accept – a major cardiac or thoracic operation. Second, the outcomes of those operations are startlingly poor, with 25% and 38% of the DNR thoracic and cardiac cohorts, respectively, dying in the hospital. Third, DNR status remains an independent risk factor for perioperative mortality when controlling for age, procedure, race, insurance status, and major comorbidities.

Maxwell et al. (2014), *PeerJ*, DOI 10.7717/peerj.245

**Table 1** Characteristics of cardiac and thoracic DNR cohorts and matched controls.

| | Thoracic cohort | | | | | Cardiac cohort | | | | |
| --- | --- | --- | --- | --- | --- | --- | --- | --- | --- | --- |
| | DNR group $n = 3{,}129$ | | Control group $n = 12{,}290$ | | | DNR group $n = 2{,}678$ | | Control group $n = 10{,}670$ | | |
| | *n* | % | *n* | % | *p* | *n* | % | *n* | % | *p* |
| Age (mean ± SD; years) | 73.8 | ±13.6 | 73.9 | ±12.8 | 0.64 | 71.6 | ±16.3 | 71.4 | ±15.9 | 0.47 |
| White | 2,090 | (66.8%) | 7,681 | (62.5%) | 0.0011 | 1,665 | (62.2%) | 5,554 | (52.1%) | 0.0018 |
| Insured | 2,756 | (88.1%) | 10,675 | (86.9%) | 0.073 | 2,361 | (88.2%) | 9,458 | (88.6%) | 0.48 |
| | | | | | | | | | | |
| Anemia | 508 | (16.2%) | 1,567 | (12.8%) | <0.0001 | 297 | (11.1%) | 1,019 | (9.6%) | 0.018 |
| Arrhythmia | 498 | (15.9%) | 1,560 | (12.7%) | <0.0001 | 620 | (23.2%) | 2,275 | (21.3%) | 0.041 |
| Coronary artery disease | 429 | (13.7%) | 1,708 | (13.9%) | 0.82 | 942 | (35.2%) | 3,708 | (34.8%) | 0.68 |
| Congestive heart failure | 353 | (11.3%) | 1,157 | (9.4%) | 0.0019 | 520 | (19.4%) | 1,707 | (16.0%) | <0.0001 |
| Chronic kidney disease | 183 | (5.8%) | 427 | (3.5%) | <0.0001 | 155 | (5.8%) | 353 | (3.3%) | <0.0001 |
| Chronic lung disease | 634 | (20.3%) | 2,437 | (19.8%) | 0.60 | 353 | (13.2%) | 1,243 | (11.6%) | 0.030 |
| Diabetes mellitus | 374 | (12.0%) | 1,540 | (12.5%) | 0.40 | 446 | (16.7%) | 1,737 | (16.3%) | 0.64 |
| Prior stroke | 39 | (1.2%) | 124 | (1.0%) | 0.24 | 71 | (2.7%) | 266 | (2.5%) | 0.63 |
| Hypertension | 952 | (30.4%) | 3,863 | (31.4%) | 0.28 | 979 | (36.6%) | 3,862 | (36.2%) | 0.74 |
| Malignancy | 927 | (29.6%) | 2,647 | (21.5%) | <0.0001 | 170 | (6.3%) | 416 | (3.9%) | <0.0001 |

**Notes.**

DNR, do-not-resuscitate; SD, standard deviation.

**Table 2 Univariate analysis of outcomes in cardiac and thoracic DNR cohorts compared to matched controls.**

| *Thoracic* | DNR n = 3,129 | | Control n = 12,290 | | | | |
|---|---|---|---|---|---|---|---|
| | *n* | *%* | *n* | *%* | OR | 95% confidence interval | *p* |
| Primary outcome | | | | | | | |
| In-hospital death | 795 | (25.4%) | 787 | (6.4%) | 4.98 | (4.47, 5.55) | <0.0001 |
| Resource utilization | | | | | | | |
| Length of stay (mean ±SD; days) | 9.4 | ±10.6 | 8.1 | ±8.7 | | | <0.0001 |
| Discharge to new SNF | 496 | (15.9%) | 1,433 | (11.7%) | 1.43 | (1.28, 1.59) | <0.0001 |
| Secondary outcomes | | | | | | | |
| Acute Kidney Injury | 71 | (2.3%) | 241 | (2.0%) | 1.16 | (0.89, 1.52) | 0.28 |
| Myocardial infarction | 46 | (1.5%) | 94 | (0.8%) | 1.94 | (1.36, 2.76) | 0.0002 |
| New congestive heart failure | 14 | (0.5%) | 62 | (0.5%) | 0.89 | (0.50, 1.59) | 0.78 |
| New stroke | 11 | (0.4%) | 29 | (0.2%) | 1.49 | (0.74, 2.99) | 0.24 |
| Respiratory failure | 154 | (4.9%) | 307 | (2.5%) | 2.02 | (1.66, 2.46) | <0.0001 |

| *Cardiac* | DNR n = 2,678 | | Control n = 10,670 | | | | |
|---|---|---|---|---|---|---|---|
| | *n* | *%* | *n* | *%* | OR | 95% confidence interval | *p* |
| Primary outcome | | | | | | | |
| In-hospital death | 1,003 | (37.5%) | 1,194 | (11.2%) | 4.75 | (4.31, 5.25) | <0.0001 |
| Resource utilization | | | | | | | |
| Length of stay (mean ±SD; days) | 12.2 | ±16.1 | 10.3 | ±13.8 | | | <0.0001 |
| Discharge to new SNF | 324 | (12.1%) | 1379 | (12.9%) | 0.93 | (0.82, 1.06) | 0.25 |
| Secondary outcomes | | | | | | | |
| Acute Kidney Injury | 264 | (9.9%) | 726 | (6.8%) | 1.50 | (1.29, 1.74) | <0.0001 |
| Myocardial infarction | 89 | (3.3%) | 242 | (2.3%) | 1.48 | (1.16, 1.90) | 0.0025 |
| New congestive heart failure | 77 | (2.9%) | 212 | (2.0%) | 1.46 | (1.12, 1.90) | 0.006 |
| New stroke | 56 | (2.1%) | 101 | (1.0%) | 2.24 | (1.61, 3.11) | <0.0001 |
| Respiratory failure | 354 | (13.2%) | 862 | (8.1%) | 1.73 | (1.52, 1.98) | <0.0001 |

**Notes.**

DNR, do-not-resuscitate; SD, standard deviation; OR, odds ratio; SNF, skilled nursing facility.

Prior analyses from the American College of Surgeons National Surgical Quality Improvement Project have shown a high postoperative mortality rate in general surgical patients who are DNR (*Speicher et al., 2013*) and have suggested that excess mortality is due to a decreased willingness to pursue aggressive interventions in the postop period (*Scarborough et al., 2012*), described as "failure to pursue rescue". While this retrospective, observational study is unable to confirm the etiology of excess mortality in the DNR groups, the resource utilization implications of this hypothesis are profound. Tremendous financial and operational resources (including the labor of surgical, anesthesia, perfusion,

**Table 3 Multivariate logistic regression models for in-hospital mortality.**

| | Thoracic | | | Cardiac | | |
|---|---|---|---|---|---|---|
| | OR | 95% confidence interval | p | OR | 95% confidence interval | p |
| DNR | 6.11 | (5.37, 6.94) | <0.0001 | 4.78 | (4.21, 5.41) | <0.0001 |
| White | 0.87 | (0.75, 1.01) | 0.066 | 1.04 | (0.90, 1.21) | 0.61 |
| Insured | 1.29 | (1.05, 1.58) | 0.017 | 1.18 | (0.96, 1.44) | 0.11 |
| Anemia | 1.21 | (1.00, 1.47) | 0.052 | 1.00 | (0.79, 1.26) | 0.99 |
| Arrhythmia | 1.53 | (1.26, 1.86) | <0.0001 | 1.12 | (0.96, 1.32) | 0.16 |
| Coronary artery disease | 1.00 | (0.82, 1.23) | 0.971 | 1.41 | (1.19, 1.67) | <0.0001 |
| Congestive heart failure | 1.24 | (0.98, 1.56) | 0.070 | 1.32 | (1.09, 1.60) | 0.0039 |
| Chronic kidney disease | 3.13 | (2.37, 4.12) | <0.0001 | 5.15 | (3.85, 6.87) | <0.0001 |
| Chronic lung disease | 1.19 | (1.00, 1.42) | 0.046 | 1.32 | (1.07, 1.64) | 0.010 |
| Diabetes mellitus | 1.05 | (0.85, 1.30) | 0.66 | 0.91 | (0.74, 1.11) | 0.33 |
| Prior stroke | 1.53 | (0.87, 2.71) | 0.14 | 0.78 | (0.50, 1.22) | 0.28 |
| Hypertension | 0.64 | (0.54, 0.76) | <0.0001 | 0.49 | (0.41, 0.58) | <0.0001 |
| Malignancy | 0.93 | (0.80, 1.09) | 0.37 | 0.87 | (0.64, 1.17) | 0.35 |

**Notes.**

DNR, do-not-resuscitate; OR, odds ratio.

nursing, and operating room technical staff, equipment and medication costs, postoperative intensive care and supportive services) are devoted to the types of cardiothoracic operations used to define our study cohorts. If "failure to pursue rescue" after making the decision to undergo a major surgical intervention plays a role in explaining the substantial elevation in postoperative mortality in this population, we believe it suggests unwise resource utilization. Perhaps even more importantly, thousands of DNR patients may be exposed to the discomfort and risk of highly invasive procedures that have a diminished prospect of a good outcome if they are not coupled with a range of certain aggressive postoperative interventions.

The major limitations of this analysis are the well-established limitations of a retrospective administrative database analysis, principally that of classification error (*Jollis et al., 1993*; *Koch et al., 2012*). The advantage of the PDD definition of DNR is that it avoids miscounting surgical patients who were not DNR at the time of admission or their operation but became DNR later in their hospital stay (e.g., after multiple postoperative complications). It remains possible that the PDD undercounts some patients who were DNR at the time of surgery (e.g., DNR order written on hospital day three, surgery on hospital day five) but if present, this classification error would only mitigate the relationship we observed.

Outcome information is similarly limited in an administrative database. Nonfatal complications are based on diagnosis codes, which do not include information on outcome severity, and are more likely to be undercounted (errors of omission) than overcounted,

because of limited reliability of diagnostic codes for perioperative complications. Our decision to define inpatient complications using the absence of the "present on admission" variable may also result in undercounting of clinically meaningful deterioration – for example, a patient with compensated heart failure present on admission whose heart failure decompensates in the perioperative period would not be defined as "new CHF" in our methodology. We suspect these phenomena, among others, help explain the observation that death was more common than any of the secondary outcome measures in both DNR cohorts. For these reasons, we used mortality as the primary outcome, as we felt it represented a more reliable endpoint.

The variables used for matching were selected to create a comparison cohort that would represent a face value "peer group" for the study cohort, but matching is likely an imperfect strategy for risk adjustment. Multivariate analysis facilitated adjustment for many other potential confounders, but the observational nature of the study implies the possibility of residual unmeasured confounders. We would qualify the notion that DNR status "independently" confers worse prognosis after cardiothoracic surgery by pointing out that a DNR order itself is unlikely to be causally harmful. Our results do not demonstrate any direct effect of DNR status on outcomes, such as a difference in quality of care. DNR status, however, might be viewed as conferring increased risk precisely because it functions as a proxy for unmeasured or unquantifiable variables that affect perioperative risk.

DNR status may capture other elements of comorbidity and poor reserve that traditional measures of comorbidity (e.g., Charlson index) do not, such as frailty (*Farhat, Velanovich & Falvo, 2012*; *Sündermann, Dademasch & Praetorius, 2011*; *Afilalo, Mottillo & Eisenberg, 2012*). The public health, operational, and public policy utility of noting DNR status as a marker of surgical risk might lie in its ease of use: unlike comorbidity scores or frailty indexes, no effort, definition, or calculation is required to determine that a patient has decided to accept DNR status. It is plainly written in the medical chart, typically in a high-profile location. If one knows only a few details about a patient, one often knows their code status.

Of course, many patients without an active DNR order might benefit from efforts to improve the quality and frequency of advance directive discussions prior to any proposed surgical intervention and discussions of risk. The absence of a DNR order in a patient whose wishes would be consistent with one is unlikely to be protective. But the converse appears to remain true: patients who have had that discussion and decided against a course of aggressive resuscitation are easily identified as extremely high-risk surgical candidates.

Further investigations will help more clearly elucidate whether there is a causal relationship between DNR status and surgery, as well as help define the subsets of DNR patients that might be better identified as comparatively good operative candidates, for the purposes of improving counseling for patients and families and framing discussions of resource utilization and the ethical dimensions of end-of-life care.

### Funding

Publication of this article was funded in part by the Open Access Promotion Fund of the Johns Hopkins University Libraries. The funder had no role in study design, data collection and analysis, decision to publish, or preparation of the manuscript.

### Grant Disclosures

The following grant information was disclosed by the authors:
Open Access Promotion Fund of the Johns Hopkins University Libraries.

### Competing Interests

The authors have no competing interests.

### Author Contributions

- Bryan G. Maxwell and Jim K. Wong conceived and designed the experiments, performed the experiments, analyzed the data, wrote the paper.
- Robert L. Lobato conceived and designed the experiments, wrote the paper.
- Molly B. Cason wrote the paper.

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
