# Peer review of "Perioperative morbidity and mortality of cardiothoracic surgery in patients with a do-not-resuscitate order"

_PeerJ, doi:10.7717/peerj.245_

## Round 0.1 · accepted · Accept

Dear Authors,Peer Reviewer two suggested DNR should be looked at in context of medical or non-medical reasons; reasons for DNR are usually recorded as this can be a medico-legal issue. Medical DNR will definitely have a poorer outcome and this is what we are looking at - in contrast if the DNR is due to religious or personal choice.I wonder this could be included in the discussion section.

Reviewer 1 ·

Basic reporting

No comments

Experimental design

The research question were clearly defined,hypothesis has been mentioned.
Set back - definition for DNR was done during 24 h of admission, those DNR status after that considered as negative, if these group of patients were included,it could possibly alter the percentage but may not give a different final end points ie DNR associated with increased In-hospital mortality

Validity of the findings

The data is quite valid and the discussion are understandable
Some speculation by the author (line 136)
Limitations of the study were stated (line 138-146)

Additional comments

Topics very relevant in term of confirming our assumption that the patients with DNR are associated with increase rate of in-hospital mortality and morbidity

Reviewer 2 ·

Basic reporting

No comments

Experimental design

Reasons for DNR should be elicited - whether it is by clinical or non-clinical grounds (e.g.faith, personal)

Validity of the findings

DNR should be looked at in context of medical or non-medical reasons; reasons for DNR are usually recorded as this can be a medico-legal issue. Medical DNR will definitely have a poorer outcome and this is what we are looking at - in contrast if the DNR is due to religious or personal choice.

·

Basic reporting

No Comments

Experimental design

No Comments

Validity of the findings

No Comments

Additional comments

Very well done retrospective analysis and interesting results with significant implications, especially in the preoperative period.